# Conservation of Host, Translocation of Parasites—Monitoring of Helminths during Population Reinforcement of the European Ground Squirrel (*Spermophilus citellus*)

Maria Kachamakova *[ID], Yasen Mutafchiev [ID], Pavel N. Nikolov and Yordan Koshev [ID]

Institute of Biodiversity and Ecosystem Research, Bulgarian Academy of Sciences, Tzar Osvoboditel Blvd 1, 1000 Sofia, Bulgaria
* Correspondence: maria.n.kachamakova@gmail.com

**Abstract:** Gastrointestinal helminth parasites can be transferred during conservation translocations and impact their outcome. The current study applied non-invasive coprological sampling to investigate the helminth infection rates and dynamics in translocated and resident European ground squirrels, during and after a population reinforcement. The FLOTAC method was calibrated and applied for the first time for the target species. In the studied coprological samples, helminth eggs belonging to Acanthocephala and Nematoda were found; the latter were morphologically identified as belonging to the families Capillariidae (Enoplida) and Trichostrongylidae (Strongylida) and superfamily Spiruroidea (Spirurida). The overall helminth prevalence and their diversity were higher in the donor colony compared to the resident one before the reinforcement. Pronounced seasonal dynamics in the parasite prevalence and diversity were observed, and their values were considerably lower in spring than in summer in both translocated and resident hosts. A year after the start of the translocation, the helminth prevalence and number of species detected in the reinforced colony had increased significantly. This is in accordance with epidemiological models and other empirical studies that predict a positive relationship between the population density of a host and the prevalence and species richness of parasites.

**Keywords:** souslik; wild rodent; FLOTAC; Acanthocephala; Capillariidae; Trichostrongylidae; Spiruroidea

## 1. Introduction

Small populations, often characterized by low levels of genetic diversity, are vulnerable to stochastic factors such as disease and pathogens. In particular, helminth parasites are known to negatively affect mammals' body condition [1] and reproductive success [2]. This is why conservation practitioners and scientists have long been concerned about the deleterious impact of pathogens during conservation actions such as captive breeding or translocations (reinforcements, reintroductions, and introductions) [3–7].

Fecal flotation techniques are recognized as a reliable and effective non-invasive method for parasite diagnostics and used by veterinarians and conservationists [8,9]. FLOTAC is a contemporary fecal flotation method that allows both qualitative and quantitative analyses [9,10]. The FLOTAC technique has already been applied for the screening of parasites in pet rodents such as guinea pigs (*Cavia porcellus*), squirrels (*Callosciurus finlaysonii*, *Callosciurus prevosti*, *Tamias striatus*, *Tamias sibiricus*, *Sciurus carolinensis*), hamsters (*Phodopus campbelli*, *Mesocricetus auratus*), chinchillas (*Chinchilla lanigera*), and murids (*Rattus norvegicus*, *Mus minutoides*) [11,12], but has not been used for free living rodents.

The European ground squirrel, *Spermophilus citellus* (L.), is a diurnal hibernating rodent belonging to the family Sciuridae. The species lives in colonies and mainly feeds on seeds and other parts of plants. Insects are also an important source of proteins and fats supporting the animals during demanding physiological events, such as hibernation

and reproduction [13]. The European ground squirrel inhabits a limited area in Central and South-Eastern Europe between the Czech Republic, the European part of Turkey, and eastern Ukraine. Bulgaria is located in the core of the species' range and has some of the largest colonies, with the highest genetic diversity [14]. The species was considered a pest until recently and serious concerns about its future survival have only arisen during the last decade [15]. Currently, the species remains unprotected by Bulgarian legislation, despite its population decline [16,17].

The parasites of *S. citellus* have been understudied. There have only been a few studies published on the fauna of its protozoan [18,19] and helminths [20–23], and only one provided ecological details [24].

The present helminthological study was conducted along with a conservation reinforcement of a small isolated population of the European ground squirrel. In accordance with epidemiological models and the contact-rate hypothesis, we expected that as the host population and density were higher in the donor colony, the parasite richness and prevalence there would also be higher [25]. Thus, our aim was to study if the parasites were co-translocated during the population reinforcement and became established in the reinforced population.

## 2. Materials and Methods

### 2.1. Reinforcement Process

Between 2017 and 2019, a population reinforcement was applied to a colony of European ground squirrels in South-Eastern Bulgaria (42.151 N; 27.006 E, 300 m a.s.l., located in a Natura 2000 site). The colony is situated near the village of Momina tsarkva, Yambol region, and had an initial number of only 20 individuals. The reinforcement was part of a LIFE+ project managed by the Bulgarian Society for Protection of Birds (BSPB) [26] and implemented in partnership with local farmers, to ensure long-term habitat management through livestock grazing. A larger, denser (estimated density in 2016: 12.95 holes/0.05 ha and abundance of up to 1000 individual [27]), and more genetically diverse ground squirrel colony in Bulgaria was used as a donor [14]. This colony is located 75 km north of the reinforcement site, near the village of Topolchane. The land of the colony has a low level of protection, and we witnessed progressive destruction of the pasture through ploughing over the course of the current translocation (2017–2019), which finally turned into a rescue action. The translocation activities were carried out in accordance with the ethical recommendations and Guidelines for Reintroductions and Other Conservation Translocations of IUCN (IUCN/SSC 2013). The animals were trapped, measured, and marked following Kachamakova et al. [26]. There were three release sessions in July 2017 (96 individuals), three in June–July 2018 (71 individuals), and two in July 2019 (46 individuals). In total, 213 animals were translocated [28].

### 2.2. Fecal Samples Collection

Fecal samples were collected from the translocated individuals during their initial capture at the donor colony. In addition, 10 individuals from the reinforced colony near the village of Momina tsarkva were captured and sampled before the release of the translocated animals. After the release, recapture sessions were organized monthly for 3–4 days during the active season (April–September) of the ground squirrels, from July 2017 to September 2019 (Table 1). During these sessions, both resident and translocated individuals were sampled, in order to study the dynamics of helminth load, in terms of the proportion of infected hosts (prevalence). Hereafter, we refer to the individuals born in the colony at Momina tsarkva before or after the reinforcement as "resident", and to the individuals brought from the colony near Topolchane as "translocated".

**Table 1.** Number of resident and translocated individuals captured at each session in the colony of Momina tsarkva. The sessions in June and July mainly included resident juveniles that emerged from the burrows for the first time at the end of May. * This session was performed with the primary aim of removing radio collars from the translocated animals, which is why the translocated individuals predominated.

| | Number of Captured Individuals | |
| --- | --- | --- |
| **Recapture Sessions** | **Resident** | **Translocated** |
| 2017: Jul | 13 | 12 |
| 2017: Aug | 3 | 9 |
| 2017: Sep | 2 | 6 |
| 2018: Apr | 2 | 5 |
| 2018: May | 4 | 6 |
| 2018: Jun | 18 | 1 |
| 2018: Jul | 22 | 2 |
| 2018: Aug | 17 | 12 |
| 2018: Sep | 7 | 2 |
| 2019: Apr | 7 | 5 |
| 2019: May | 9 | 3 |
| 2019: Jun | 13 | 0 |
| 2019: Jul | 11 | 0 |
| 2019: Sep * | 1 | 7 |

The fecal samples collected during the handling of the animals were immediately fixed in 70% ethanol and stored in a refrigerator until processing. It should be noted that ethanol was found to be a suboptimal fixative for coprological studies of helminths, but that a good level of recovery is observed when the flotation solution is appropriately chosen [29].

*2.3. FLOTAC Calibration and Processing*

The FLOTAC method was used for detection of the parasitic elements [30]. This technique has not previously been applied for coprological samples of *Spermophilus* spp., and calibration was applied, following the protocol by Cringoli [31]. For that purpose, a part of the fecal samples were mixed together and used to test the flotation performance of 9 solutions prepared as described by Cringoly [31]. For each solution, 6 replicates were performed. Zinc sulfate solution with a 1.35 gravity was selected based on the number of floating helminth eggs (see Table 2). Each sample was processed according to the FLOTAC DUO technique [31] and studied under a compound light microscope Olympus BX 41, Japan.

**Table 2.** Flotation solutions and their performance. The most appropriate solution is marked in bold.

| Flotation Solution | Specific Gravity | Floating Parasitic Elements (Mean ± SE) |
| --- | --- | --- |
| Sucrose and formaldehyde | 1.2 | 6.8 (±5) |
| Sodium chloride | 1.2 | 0.3 (±0.2) |
| Zinc sulphate | 1.2 | 0 |
| Sodium nitrate | 1.2 | 0 |
| Sucrose and potassium iodomercurate (Rinaldi's solution) | 1.25 | 0 |
| Magnesium sulphate | 1.28 | 11 (±2) |
| **Zinc sulphate** | **1.35** | **30 (±4)** |
| Potassium iodomercurate | 1.44 | 1.6 (±0.4) |
| Zinc sulphate and potassium iodomercurate | 1.45 | 0.2 (±0.2) |

In order to enable quantitative comparison of the results, the weight of each sample was noted before processing (range: 0.09–0.57 g, mean: 0.36 g). The parasite eggs were

identified by comparison with those found in female helminths identified at species level and deposited as voucher and collection materials in the Parasite Collection of the Institute of Biodiversity and Ecosystem Research, BAS. Based on the number of eggs counted and the sample weight, the quantity of eggs per gram was calculated.

## 2.4. Helminthological Examination

Eight ground squirrels found dead during the field work in the area of the donor colony (4 translocated and 4 resident) were subjected to thorough helminthological examination.

## 2.5. Statistical Analyses

We used the prevalence (the number of infected hosts divided by the number of examined animals) to compare the helminth infection in different groups of individuals based on their origin, age, sex, and the time of sampling throughout the active season. The prevalence is a proportion; therefore, a binomial generalized model was used. For that purpose, the data were combined based on the season, in order to achieve larger sample sizes and to check the effect of sex, age, origin, and year. Models were designed for the total prevalence and for the prevalence of each helminth taxa separately in the coprological samples. Acanthocephalan and capillariid eggs were only found in 7 and 5 animals, respectively; therefore, no further statistical analyses were performed for these two groups.

In order to acquire a standardized variable for the quantitative analysis, we counted the number of helminth eggs in each sample and divided this number by the sample weight in grams. The resulting variable was eggs per gram (EPG). Over-dispersed data were analyzed with quasi-Poisson generalized linear models, to investigate the impact of the explanatory factors: origin, year, sex, age, etc. (Table 3). Only infected individuals were included in the models, and these were built separately for the total EPG and for each helminth group when the sample size was higher than 15.

The final models for prevalence and EGP were obtained after the backward selection of non-significant terms, until only the significant ones were left. In order to simplify the model parameters, the animals were placed into two age classes: juveniles (born in the year of sampling) and adults (more than one year old). All analyses were performed in R-software (Version 4.2.2—31 October 2022).

**Table 3.** Parameters of the statistical analyses applied with prevalence as a response variable. Glm—generalized linear model; T—translocated; R—resident.

| | Groups | Periods | N | Response Variable | Explanatory Variables | Model | Significant and Nearly Significant Variables |
|---|---|---|---|---|---|---|---|
| 1 | T + R | 2017: Jul | 30 | Prevalence—all helminths | Sex, age, origin | Binomial glm | |
| 2 | R | 2017: Jul–Aug 2018: Jul–Aug | 33 | Prevalence—all helminths | Year, sex, age | Binomial glm | **Year:** z = 3.027, $p = 0.002$ (2018 > 2017) |
| 3 | T | 2017: Jul 2018: Jun–Jul | 32 | Prevalence—all helminths | Sex, age, year | Binomial glm | – |
| 4 | T | 2017: Jul 2018: Jun–Jul | 32 | Prevalence—Spiruroidea | Sex, age, year | Binomial glm | **Year:** z = 2.171, $p = 0.030$ (2017 > 2018) **Sex:** z = −2.137, $p = 0.033$ (m < f) **Age:** z = −1.747, $p = 0.080$ (juv < ad) |
| 5 | T | 2017: Jul 2018: Jun–Jul | 32 | Prevalence—Trichostrongylidae | Sex, age, year | Binomial glm | **Year:** z = −1.945, $p = 0.05$ (2017 > 2018) |

**Table 3.** *Cont.*

| | Groups | Periods | N | Response Variable | Explanatory Variables | Model | Significant and Nearly Significant Variables |
|---|---|---|---|---|---|---|---|
| 6 | T + R | 2017: Aug–Sep 2018: Aug–Sep 2019: Aug–Sep | 47 | Prevalence—all helminths | Sex, age, year, month, origin | Binomial glm | – |
| 7 | T + R | 2017: Aug–Sep 2018: Aug–Sep 2019: Aug–Sep | 47 | Prevalence—Spiruroidea | Sex, age, year, month, origin | Binomial glm | **Origin:** $z = 2.221$, $p = 0.026$ (T > R) |
| 8 | T + R | 2017: Aug–Sep 2018: Aug–Sep 2019: Aug–Sep | 47 | Prevalence—Trichostrongylidae | Sex, age, year, month, origin | Binomial glm | – |
| 9 | T + R | 2018: Apr–May 2019: Apr–May | 35 | Prevalence—all helminths | Sex, year, origin | Binomial glm | **Origin:** $z = 2.201$, $p = 0.028$ (T > R) |

## 3. Results

### 3.1. Helminth Diversity

The eight dissected ground squirrels were free of nematodes, trematodes, and cestodes; however, in the small intestines of four animals (3 translocated and 1 resident) the acanthocephalan *Moniliformis moniliformis* (Bremser, 1811) (Moniliformida: Moniliformidae) was found. In the coprological samples collected from live animals during the study, four types of helminth eggs belonging to Acanthocephala and Nematoda were detected. The acanthocephalan eggs found in the dissected ground squirrels were 75–88 × 40–48 μm (average 88 × 40 μm; n = 10) in size (Figure 1A). The three types of nematode eggs were identified as belonging to the family Capillariidae (Enoplida), 52–56 × 25–29 μm (av. 55 × 26 μm; n = 10) in size (Figure 1B); the family Trichostrongylidae (Strongylida), 80–92 × 36–44 μm (av. 84 × 40 μm; n = 10) in size (Figure 1C); and the superfamily Spiruroidea (Spirurida), 37–39 × 20–30 μm (av. 39 × 21 μm; n = 10) in size (Figure 1D).

### 3.2. Dynamics of the Helminth Prevalence in Chronological Order

In July 2017, before the reinforcement, only eggs of Acanthocephala were present in 10% of the samples collected from the colony at Momina tsarkva, whereas three helminth taxa: Capillariidae, Trichostrongylidae, and Spiruroidea were found in the translocated animals, with an overall helminth prevalence of 53%. The difference in the prevalence rate between the resident and the translocated hosts during that period was statistically significant ($z = 2.266$, $p = 0.024$). There were no significant differences in the proportion of infected individuals between the sex and age classes (Table 3, line 1). Later in the season, in August and September, the helminth prevalence increased for both groups (Table 4; Figures 2 and 3).

All resident animals sampled in the spring of 2018 (April–June) were negative for helminths. Whereas, one third of the translocated animals were positive for *S. kutassi*, but eggs of other helminths were not detected.

In the second year of translocation (July 2018), the translocated animals were infected with Capillariidae, Trichostrongylidae, Spiruroidea, and Acanthocephala, and the overall helminth prevalence was 60%. Compared to 2017, the prevalence was significantly higher for Spiruroidea ($z = 2.171$, $p = 0.030$) and significantly lower for Trichostrongylidae ($z = -1.945$, $p = 0.05$) (Table 3, lines 4 and 5). In addition, females were more often infected with Spiruroidea than males ($z = -2.137$, $p = 0.033$) (Table 3, line 4).

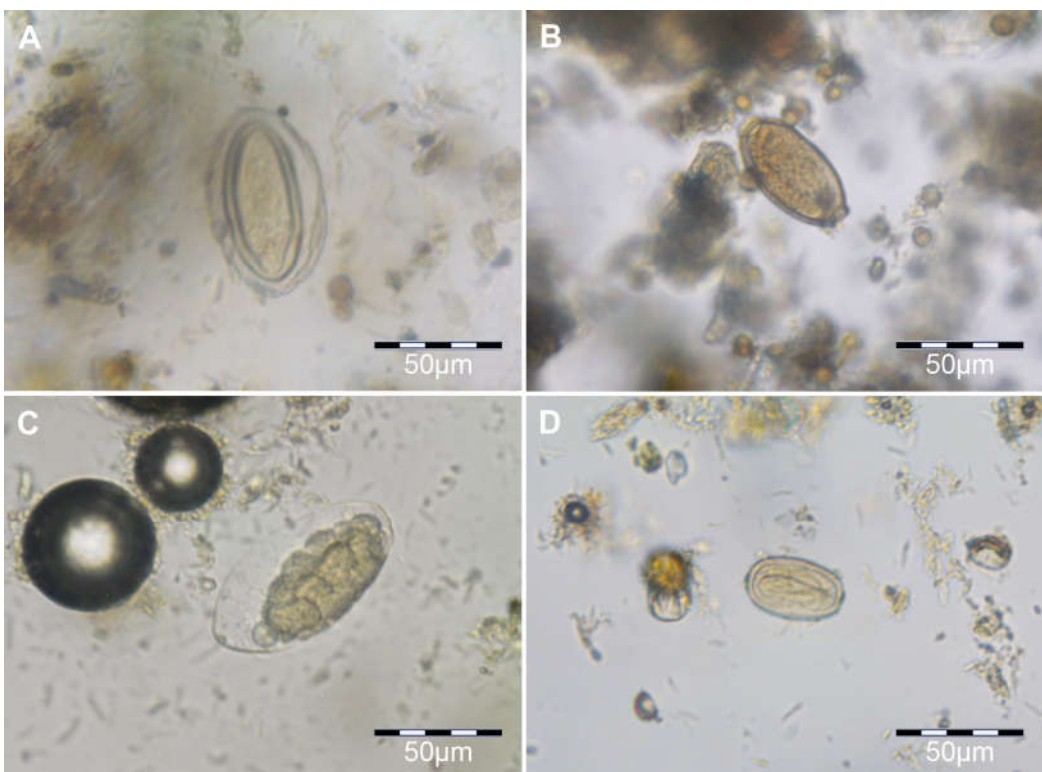

**Figure 1.** Eggs of helminths found during the coprological study, identified as belonging to Acanthocephala (**A**), Capillariidae (**B**), Trichostrongylidae (**C**), and Spiruroidea (**D**).

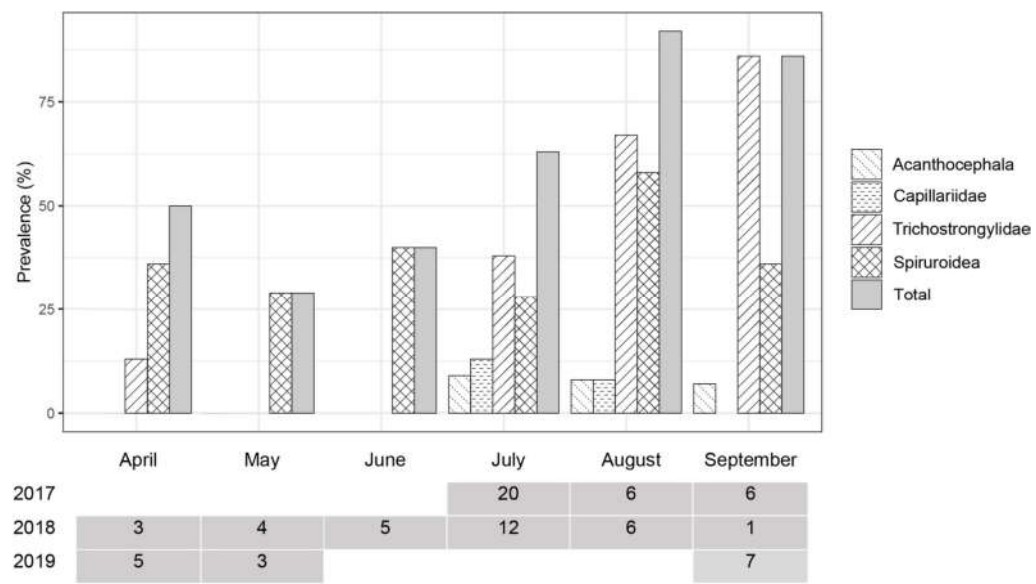

**Figure 2.** Seasonal dynamics for the studed period in the prevalence of the observed helminth taxa in the translocated individuals, summarized by month. Cells below the graph represent the period of sampling by month, with the corresponding sample sizes.

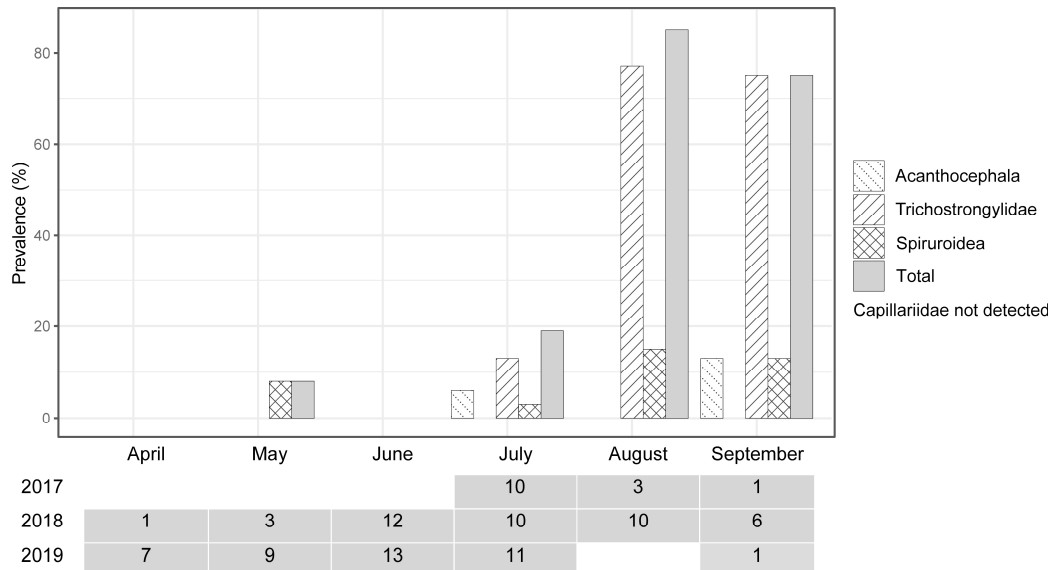

**Figure 3.** Seasonal dynamics for the studied period in the prevalence of the observed helminth taxa in the resident individuals summarized by month. Cells below the graph represent the period of sampling by month, with the corresponding sample size.

**Table 4.** Prevalence (%) of helminth taxa found in the coprological samples of resident and translocated ground squirrels. R = resident, T = translocated. Notes: The "Jul" datasets contain samples at the moment of translocation. * Only juvenile individuals were captured; therefore, the results were not included in the analysis.

| Period | Acanthocephala | | Capillariidae | | Trichostrongylidae | | Spiruroidea | | Sample Size | |
|---|---|---|---|---|---|---|---|---|---|---|
| | R | T | R | T | R | T | R | T | R | T |
| 2017: Jul | 10 | 0 | 0 | 12 | 0 | 47 | 0 | 12 | n = 10 | n = 17 |
| 2017: Aug and Sep | 0 | 10 | 0 | 10 | 25 | 80 | 0 | 70 | n = 4 | n = 12 |
| 2018: Apr, May and Jun | 0 | 0 | 0 | 0 | 0 | 0 | 0 | 33 | n = 16 | n = 12 |
| 2018: Jul | 10 | 20 | 0 | 13 | 40 | 13 | 10 | 53 | n = 10 | n = 15 |
| 2018: Aug and Sep | 6 | 14 | 0 | 0 | 88 | 57 | 19 | 43 | n = 16 | n = 7 |
| 2019: Apr, May and Jun | 0 | 0 | 0 | 0 | 0 | 38 | 29 | 13 | n = 7 | n = 8 |
| 2019: Jul * | 0 | - | 0 | - | 0 | - | 0 | - | n = 12 | n = 0 |
| 2019: Sep | 0 | - | 0 | - | 100 | - | 14 | - | n = 7 | n = 0 |

Out of the 32 individuals from Topolchane sampled at the moment of translocation (July 2017 and July 2018), one individual had a coinfection with three types of parasites (Acanthocephala, Trichostrongylidae and Spiruroidea). There were also five coprological samples containing two types of helminth egg: Acanthocephala with Trichostrongylidae (1 sample), Trichostrongylidae with Spiruroidea (1 sample), Trichostrongylidae and Capillariidae (2 samples), and Spiruroidea with Capillariidae (2 samples). Twelve translocated individuals were recaptured and resampled 24–63 days after the release. In ten of them (6 in 2017 and 4 in 2018), helminths that were absent initially were found at resampling.

As regards the overall helminth prevalence in the colony of Momina tsarkva, the difference was significant between July 2017 before the translocation (n = 10) and in the same month of the next year (n = 10), (z = 3.027, *p* = 0.002) (Table 3, line 2). While only Acanthocephala (10%) was documented in the resident hosts before the reinforcement, in the resident individuals sampled in July 2018 were found Acanthocephala (10%), Trichostrongylidae (40%), and Spiruroidea (10%).

In August and September 2018, eggs of Acanthocephala, Trichostrongylidae, and Spiruroidea were detected in both resident and translocated individuals. The prevalence was again higher than earlier in the season for Trichostrongylidae and Spiruroidea (except

for the translocated hosts), similarly to the late summer of 2017 (Figures 2 and 3). Binomial models were built based on all the samples collected in the late summer (August and September 2017, 2018, and 2019), investigating the impact of origin, sex, age, month, and year on the infection rates of each helminth type. These models revealed that Spiruroidea was present more often in the translocated than in the resident individuals ($z = 2.221$, $p = 0.026$) (Table 3, line 7). The prevalences for the other helminth groups were very similar between the residents and the translocated hosts (Table 4). The impact of the other explanatory variables (sex, age, year) was not significant for any helminth group for this period (Table 3, lines 7, 8, and 9).

During the spring of 2019, the translocated animals had eggs of Trichostrongylidae (38%) and Spiruroidea (13%), whereas only eggs of the latter species were present in 29% of the residents. When the spring samples of both 2018 and 2019 were statistically analyzed, the translocated animals were significantly more infected than the residents ($z = 2.201$, $p = 0.028$) (Table 3, line 9).

### 3.3. Seasonal, Age-, and Sex-Related Variations in Helminth Prevalence

Seasonal dynamics in the proportion of infected individuals were observed in all age and origin groups (Table 5; Figures 2 and 3), with helminth taxa showing a lower prevalence in spring and gradually increasing through to the end of the summer.

**Table 5.** Prevalence (%) of helminth taxa found in the coprological samples of resident and translocated ground squirrels. Juv = juveniles, Ad = adults.

| Month | Acanthocephala | | Capillariidae | | Trichostrongylidae | | Spiruroidea | | Sample Size | |
| | Juv | Ad | Juv | Ad | Juv | Ad | Juv | Ad | Juv | Ad |
| --- | --- | --- | --- | --- | --- | --- | --- | --- | --- | --- |
| Jun | 0 | 0 | 0 | 0 | 0 | 0 | 0 | 22 | n = 21 | n = 9 |
| Jul | 7 | 9 | 2 | 14 | 27 | 23 | 10 | 27 | n = 41 | n = 22 |
| Aug | 0 | 10 | 0 | 10 | 67 | 80 | 20 | 50 | n = 15 | n = 10 |
| Sep | 12 | 0 | 0 | 0 | 76 | 100 | 29 | 20 | n = 17 | n = 5 |

In June, when the juveniles were approximately 2 months old, no helminth eggs were detected in their feces (Table 5). A month later, eggs of all four helminth taxa were released by the juveniles (except for 2019). As a whole, the prevalence of helminths was higher in the adult ground squirrels than in the juveniles (Table 5).

The proportions of infected individuals among the sexes were found to be as follows: Acanthocephala (males—7%, females—3%), Capillariidae (males—4%, females—2%), Trichostrongylidae (males—32%, females—29%), and Spiruroidea (males—18%, females—19%), based on all sampled males (n = 76) and females (n = 99).

### 3.4. Variations in the EPG

Egg number per gram exhibited considerable variability for all helminth taxa: Acanthocephala (range 2–760), Capillariidae (range 8–86), Trichostrongylidae (2–65), and Spiruroidea 2–400) (Figure 4).

In total, four models were built investigating the impact of different factors on EPG (Table 6). One model showed that the individuals translocated in 2018 had more EPG than those translocated in 2017 ($z = 2.215$; $p = 0.041$) (see Table 6, line 1). Another model found that in summer the adults had more EPG compared with the juveniles ($z = -2.393$, $p = 0.021$) (Table 6, line 3).

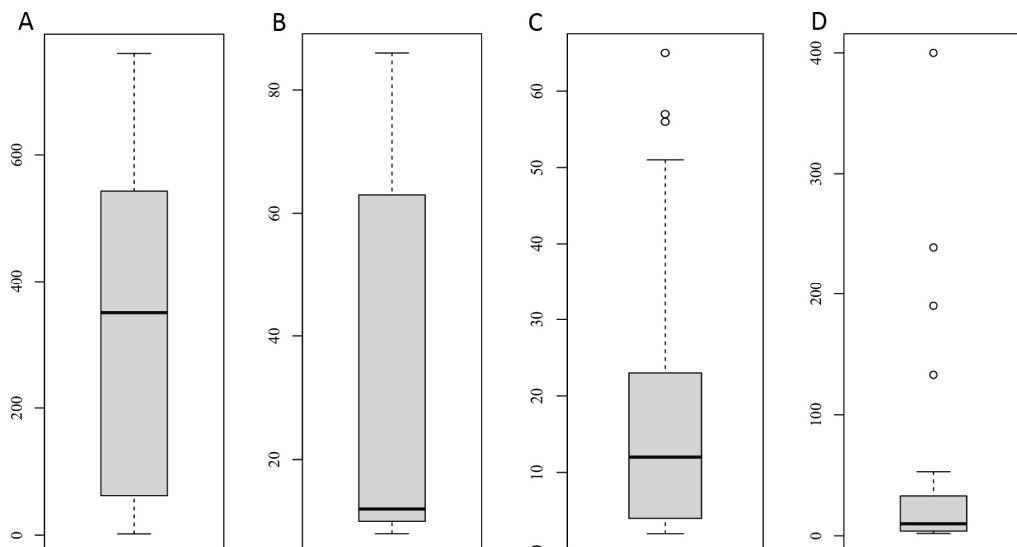

**Figure 4.** Egg count per gram of the observed helminth taxa in all samples. (**A**) Acanthocephala; (**B**) Capillariidae; (**C**) Trichostrongylidae; (**D**) Spiruroidea.

**Table 6.** Details of the statistical analyses applied with EPG as response variable. Glm—generalized linear model; T—translocated; R—resident.

| | Groups | Periods | N | Response Variable | Explanatory Variables | Model | Significant and Nearly Significant Variables |
|---|---|---|---|---|---|---|---|
| 1 | T | 2017: Jul 2018: Jun–Jul | 19 | EPG—all helminths | Sex, age, year | quasi–Poisson glm | **Year:** z = 2.215; *p* = 0.041 (2018 > 2017) |
| 2 | T + R | 2017: Aug–Sep 2018: Aug–Sep 2019: Aug–Sep | 40 | EPG—all helminths | Sex, age, year, month, origin | quasi–Poisson glm | – |
| 3 | T + R | 2017: Aug–Sep 2018: Aug–Sep 2019: Aug–Sep | 15 | EPG—Spiruroidea | Sex, age, year, month, origin | quasi–Poisson glm | **Age:** z = −2.393, *p* = 0.021 (juv < ad) |
| 4 | T + R | 2017: Aug–Sep 2018: Aug–Sep 2019: Aug–Sep | 36 | EPG—Trichostron–gylidae | Sex, age, year, month, origin | quasi–Poisson glm | – |

## 4. Discussion

### 4.1. Helminth Diversity

The four helminth taxa detected in the current study have different life cycles and host preferences. Only helminth eggs of Acanthocepha were found to be common to the donor colony and the colony at Momina tsarkva before the reintroduction. The eggs were morphologically identical to those observed in the females of *M. moniliformis* found in the dissected animals and likely belonged to this species. *Moniliformis moniliformis* was reported as a parasite of *S. citellus* in Poland [32] and Bulgaria [21]. It is a common parasite in murid rodents, but it is also known from other hosts such as cats, dogs, and foxes [33,34]. This parasite is known to have a prepatent period of 6 weeks [35]. The presence of the acanthocephalan in the studied colony near Momina tsarkva was likely not dependent of the density of the ground squirrels, as the parasite may use alternative definitive hosts such as Harting's vole and carnivores, which were present in the area of the colony.

Almost all nematodes of the family Capillariidae have a direct life cycle, although some need to be ingested by an earthworm to become infective [36]. Capillariids had not previously been reported in ground squirrels in Bulgaria, and this was the rarest of the four detected parasite taxa, being found in 3% of all samples. It is possible that the ground squirrel is an accidental host for this parasite or that the low intensity levels of infection make it hard to detect.

The trichostrongylid eggs found during the study may belong to *Trichostrongylus colubriformis* (Giles, 1892), a nematode that was found in the ground squirrel in Bulgaria by Genov [21] and Stefanov et al. [22]. This nematode with a direct life cycle is a common parasite of sheep. The ground squirrel colony at Momina tsarkva is a pasture intensively grazed by sheep and sometimes cows. The ground squirrels possibly acquired trichostrongylid nematodes by consuming grass and other plants that are well represented in their diet throughout the active season [13]. It cannot be excluded that one or more trichostrongylid species parasitizing sheep were present in the area of the ground squirrel colony before the reinforcement. The highest prevalence of these eggs in the studied samples was observed in late summer, 88% in 2018 and 100% in 2019 (Table 4). The prepatent period of trichostrongylids is about 3–4 weeks [36].

The spiruroid eggs found in the coprological samples may belong to *Streptopharagus kutassi*, a parasite of sciurid and gerbillid rodents from Eastern Europe to Central Asia [37]. *Streptopharagus kutassi* (Schulz, 1927) was reported as a parasite of *S. citellus* in Bulgaria [21,22]. The life cycle of *S. kutassi*, similarly to that of all spirurid nematodes, requires an arthropod as an intermediate host. Although little is known about the life cycle of *S. kutassi*, its infective larvae were found in the tenebrionid beetle *Mesostena angustata* (Fabricius) (reported as *Pimelia angustata*) [38]. Tenebrionids are a common part of the diet of *S. citellus* (pers. observation). There are no other appropriate final hosts aside from the ground squirrel for this nematode in the studied area. It is possible that other spiruroid nematodes, such as *Gongylonema longispiculum* Schulz, 1927 (Gongylonematidae) reported in the European ground squirrel in former Yugoslavia, may be present [23]. The prepatent period of spiruroid nematodes varies greatly, from about 3–4 weeks up to a few months [36]. In our study, spiruroid eggs were found in the feces of four 3-month-old juveniles, indicating a prepatent period shorter than 2 months. The prevalence of the spiruroid eggs remained higher in the translocated animals resampled during the subsequent year. This indicates that the worms survived winter in their definitive host. During the spring of 2019, the resident ground squirrels already exhibited a high prevalence of that nematode (Table 4), apparently already successfully circulating in the area of the colony. In addition, at that time, the host population increased as a result of the reinforcement [28] and this could also have benefited the parasite.

### 4.2. Helminth Prevalence after the Translocation

By nature, population reinforcement represents the transfer of animals from large viable populations to others with low numbers and density. Our study demonstrates that this also involves co-transfer of parasites. Before the reinforcement, the colony at Momina tsarkva was estimated at about 20 individuals [26]. We found out that the colony had a low diversity and prevalence of helminths compared to the one near Topolchane, where the estimated number was from several hundred up to more than one thousand individuals [27]. This was likely caused by the low density of the host colony, being unable to ensure successful establishment of helminth populations, and was in agreement with the contact rate hypothesis predicting a positive relationship between the host population density and the prevalence and species richness of parasites, as also confirmed by empirical studies [6,39,40]. The significant variation in the helminth prevalence between the years in the donor colony could have been related to different local factors, including demographic changes in the host population.

The colony at Momina tsarkva expanded after the reinforcement [35], despite the fact that the helminth prevalence increased significantly from 10% (in July 2017 before

the reinforcement) to 50% (in July 2018) and reached levels close to those of the donor colony (53% in 2017 and 66% in 2018). This suggests that although newly introduced, these parasites may not have necessarily had a negative impact on the host population. One possible explanation could be that the donor and the resident populations are genetically similar and would be expected to share the same parasite species, which are, after all, an inseparable part of a healthy ecosystem [41].

### 4.3. Seasonal, Age-, and Sex- Related Variations in Helminth Prevalence

A higher prevalence of parasites later in the season (August and September) was observed in both translocated and resident individuals (Tables 2 and 4; Figures 2 and 3). This was likely a result of the seasonal dynamics in the parasites life cycle and host activity. Studies have shown that hibernation, as in other sciurids, negatively affects helminth parasites [42–44]. This observation could be used to improve the planning of future helminth screening actions.

The lower helminth prevalence in the juvenile ground squirrels compared to that of the adults (Table 5) was probably due to the short time they had to acquire parasites and the prepatent period of the parasites. The higher proportion of female animals infected with spiruroid nematodes among the translocated ground squirrels could be explained by their different behavior patterns and differences in their diet. Hillegass et al. [45] also reported higher endoparasite loads in female ground squirrels than in the males. The higher consumption of invertebrates, which serve as intermediate hosts of spirurids, by females, in order to meet their nutritional needs during pregnancy and lactation, could explain this observation.

### 4.4. Variations in EPG

Studies have shown that the number of nematode eggs released is non-periodic in long runs of day-to-day records and that the relationship between the egg count and worm load is non-linear; the egg output per female helminth declines as the parasite burden increases [46,47]. This complexity could explain the large variation in the observed EPG and the lack of significant factors in half of the models. However, the significantly higher number of spiruroid EPG observed in the adults compared to the juveniles was in accordance with the trend in the helminth prevalence (discussed above).

## 5. Conclusions

A study showed that nearly 15% of conservation translocations experienced difficulties caused by diseases and parasites [48]. Thus, methods for non-invasive parasitological screening of endangered species, such as the one presented here, are important tools for ensuring the successful output of conservation actions. The fact that some of the individuals developed helminth infections after their release in an environment free or with low presence of these helminth species shows that it is likely that these translocated animals had hidden infections. Immature helminths could have been present in hosts in the moment of translocation, and these cannot be diagnosed via fecal examination. In that case, to prevent the translocation of parasites, a longer stay in quarantine would be needed; however, this is against the recommendations given for such conservation activities (IUCN/SSC 2013), as it is associated with increased stress for the animals. Last but not least, parasite conservation is a new but growing consideration among the community of conservation scientists and practitioners [49,50]. In this regard, our survey is in line with "Goal 1. Add parasite biodiversity to survey efforts for free-living species" and could be considered as a first step for achieving "Goal 7. Standardize protocols for including parasites in faunal translocations and ex-situ faunal conservation, including cost–benefit justifications" of the global parasite conservation plan [50]. The awareness of the critical role of parasites in ecosystem functioning and evolution calls for restriction of the use of anti-parasitic chemicals that disrupt the parasite–host relationship, which could cause unexpected consequences for both sides. The lack of evidence for a negative impact of the

co-translocated parasites on the reinforced colony within the frame of the present study supports this concept.

**Author Contributions:** Conceptualization, Y.K.; methodology, P.N.N. and Y.M.; formal analysis, M.K.; writing—original draft preparation, M.K.; writing—review and editing, Y.M., P.N.N. and Y.K.; visualization, M.K. and Y.M.; project administration, M.K.; funding acquisition, Y.K and MK. All authors have read and agreed to the published version of the manuscript.

**Funding:** The translocation activities were funded by the LAND for LIFE project (LIFE14 NAT/BG/001119, coordinated by the BSPB) (www.landforlife.org). Work on the manuscript development was supported by the Bulgarian Ministry of Education and Science under the National Research Programme "Young scientists and postdoctoral students" approved by DCM # 577/17.08.2018.

**Institutional Review Board Statement:** Not applicable.

**Informed Consent Statement:** Not applicable.

**Data Availability Statement:** Not applicable.

**Acknowledgments:** We give special acknowledgment to Zlatka Dimitrova (Trakia University) for identification of the specimens of *Moniliformis moniliformis*. We would like to thank Dimitra-Lida Rammou for her valuable assistance during the field work. We warmly thank Svetoslav Spasov and Dimitar Gradinarov (BSPB), without whom the translocation would not have been possible. We also thank Anton Sokolov, Svetla Todorova (Sofia University), Dimitar Ragyov (IBER-BAS), and Kristina Yonkova for their participation in the sampling process. The anonymous reviewers are acknowledged for their valuable comments on the manuscript.

**Conflicts of Interest:** The authors declare no conflict of interest.

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
