# Peer review of "Conservation of Host, Translocation of Parasites—Monitoring of Helminths during Population Reinforcement of the European Ground Squirrel (Spermophilus citellus)"

_diversity, doi:10.3390/d15020266_

Round 1
Reviewer 1 Report
Dear Authors,
Please see my review in the attached file.

Author Response
Thank you for the comments and corrections. Authors’ answers are given in italic after each paragraph of the review (Word document attached).

Reviewer 2 Report
Review report
Brief summary
The topic of the manuscript entitled “Conservation of host, translocation of parasites – results of noninvasive monitoring of the helminth infection levels during population reinforcement of the European ground squirrel” is of paramount importance for the effective conservation of the endangered European ground squirrel (EGS). Its importance is well demonstrated by two valid conclusions of the monitoring of the species in Hungary or in other countries within its distribution area: 1. A general decline has been going on in spite of many conservation actions during the last decade, 2. There are seemingly unexplainable local extinctions. Another reason to highlight the importance of the choice of the topic is the need for parasite management in animal translocations. There are several papers about this topic mentioning its importance from the perspectives of human health, animal fitness, or simply the dangers of reintroducing foreign parasites into new areas (long-term reintroduction success) but real empirical studies on wild animals are less frequent.
General comments
The manuscript is well organized and more-or-less clearly written. Cited references are relevant and reflect a sound knowledge about the topic. The study is scientifically sound and experimentally correct, therefore it could be reproduced. Regarding the statistical approach, a binomial GLM model was used to analyse the relationship between different factors including sex, age, year, sample size (could be added to the model as a predictor), etc. and the prevalence of helminths (total and each taxon separately; one total model and other ones for each taxon?).
Regarding the Introduction, I believe that being more specific in the introduction about a few concepts, such as specific epidemiological models related to the concept (contact-rate hypothesis), significance of parasite load of animals on fitness, survival particularly after translocations, etc. should be discussed in the introduction to some extent to put the study in a bit broader context and give the reader a more comprehensive picture on the topic tightly relevant to the study. Moreover, I would suggest to formulate hypotheses about the temporal dynamics of abundance and diversity of helminths in accordance with the epidemiological models discussed in the introduction. Being more specific in the introduction about the expected outcome would improve the paper.
Regarding the statistics (Methods and Results), I would be glad to see more results (e.g. a table or in the text) about the model parameter estimates, the model adequacy (model diagnostics), etc. Besides, I miss the details of model building, such as the choice of the link function (probit, logit, etc.). All in one, more details would be helpful in the Methods to understand the methodological-statistical approach and more structured communication of the results, certainly, in accordance with the methods would help the readers see the solid scientific approach of the questions. Otherwise, conclusions are well related to the results and arguments, they are appropriately formulated.
Although it perhaps sounds much work for the first reading I would add few (1-3) sentences for each idea or concept to be mentioned (Intro), therefore it would not require a huge amount of work. Regarding the statistics, I think that not much is missing to reach better explained model building, selection and interpretation in accordance to the improved methods.
Overall, I would suggest to revise the current version before publication.
A few specific comments, here-below, will provide details or suggestions on how to improve the manuscript.
Specific comments
Introduction
Line 18 helminth or helminths prevalence – choose and stay consistent (Abstract);
Methods
L69-70 “…biggest, denser…”-- Density? Population size? Some estimations about the abundance (size and density) and area of occurrence of the donor colony would be useful to read about;
Line 77 Data about the number of individuals translocated during the three distinct translocation events would be welcomed;
L82-83 I would recommend to start that sentence with “After each release, recapture sessions were ...”; How many individuals, either resident or translocated, were recaptured during those trapping sessions?
L107-109 I suggest this sentence instead of the original in the text “For the needs of proper identification of the circulating helminths, the eight ground squirrels found dead during the field works in the donor (? animals) and resident colonies (? animals) were subjected to thorough helminthological examination.”;
L114 ”…was….” Instead of were;
L121-125 That part should be explained more. What are the conceptual and practical relationships between prevalence or parasitic load and the number of eggs per gram per host? Besides, as that number does not reflect parasitic load then what indicator was eventually used in the analysis? Anyway, it was confusing to me and would require some clarification about the relationship between the parasitic load, prevalence, and egg number per gram;
L129-136 That sentence is too long and difficult to read. Please rephrase it;
L212 “…dynamicS….” I think it is rather used in plural;
Figure 3. It is difficult to distinguish the different taxa and columns due to colouring-shades. I would suggest you to use different textures or colours () instead.
Table 3. Legend contains mistakes.
Discussion and Conclusions
L294 A sentence about the effect of hibernation on helminths and as a result perhaps on the survival of EGS in different parts of the year (hibernation, active season) or body, physical condition, fitness would add a broader perspective on the importance of the study. If there is no data about this, then perhaps data on another species or group like ground dwelling sciurids would still be useful to read;
L328 Is there a way of decreasing or eliminating helminths in EGS by anti-helminthic drugs? It could prevent the unnecessary reintroduction of helminths into resident colonies? To talk about this matter would perhaps be interesting to conservationist, practitioners, etc.

Author Response

(The authors gave the same response as above.)

Reviewer 3 Report
The manuscript presents very interesting results about the presence/prevalence of GI parasites in translocated and resident specimens of the European ground squirrel. Parasitic diseases are useful to study adaptation to a new environment, not only for their presence but also for the possibility of studying the responses of translocated/resident hosts to a new species and, therefore, to evaluate the success or failure of a translocation.
I have some comments that the authors should incorporate into the manuscript.
Underlined in yellow are some sentences that need to be rewritten to make them understandable.
The title is too long but should include the scientific name of the European ground squirrel.
Methodology: Describe how parasite eggs were characterized, including taxonomic keys, description of morphology, eggs size. Describe the organ where M. moniliformes specimens were found,any macroscopic alteration, size of the adult parasites, sex, photos of the parasites.
Regarding sampling times, the authors should find a way to make them more understandable. Instead of dates, months or number of weeks in order to understand possible prepatent periods of the parasites. The authors should explain why they used FLOTAC and not another flotation technique.
What are the prepatent periods of the parasites? That could support the findings in adults and juveniles. The authors should describe the parasites found, their life cycles, some details as if they are specie- specific.
What is the diet of the EGS that allowed them to acquire Trychostrongylus colubriformes from the sheep pastures? If EGS have a varied diet such as seeds, insects and grass, was there any seasonal variation that compelled these animals to consume grass rather than another diet? On the other hand, sheep prefer leaves, stems, and moisture-containing feedstuffs. Why T. colubriformes and not another nematode from sheep if sheep generally have polyparasitism? Any report about that?
The authors should present another image of T. colubriformes egg. I am not so sure that T colubriformes was the parasite mentioned by the authors. The authors must present any other proof of parasitic species found in samples. The dead specimens did not necessarily have to be parasitized with the same species as the resident and translocated living specimens.
Please review the references according to MDPI standards
Finally, I suggest the authors develop thoroughly both the introduction and discussion.

Author Response

(The authors gave the same response as above.)

Round 2
Reviewer 1 Report
Dear Authors,
Thank you for taking into consideration my suggestions.
I believe that the overall changes you have made to your manuscript have improved it significantly. Among others, I appreciate your point of view on the parasite conservation concept, discussed now in the conclusions.
My final minor suggestions are the following:
Title: Conservation of host / translocation of parasites / non-invasive monitoring: too many actions on a raw without any connecting words. This is a confounding title, maybe you would consider rephrasing, especially after the following comment regarding the “non-invasive” characterisation of the process.
Non-invasive: There is no apparent reason that this quality of the examinations is hyped. Is invasive monitoring applied or proposed in any (other) case? A faecal examination is a mainstream way of monitoring parasitism, nothing novel to claim here. I am sure that nobody would consider killing some animals to assess their parasites.
Line 11. Delete non-invasive. Applies faecal sampling.
Actually, I would delete non-invasive throughout the text. We are talking about faecal parasitological examinations here. These are de facto non-invasive. It is both redundant and misleading to use the characterisation non-invasive.
Lines 48-49 Please rephrase to “…of proteins and fats supporting the animals in demanding physiological events like ….”
Author Response
Dear Editor, Dear Reviewer,
Thank you for the comments and suggestions,
Title: the title was changed according to reviewer's suggestions.
Non-invasive: We consider that this is a key feature of our study taking into account the conservation aspect of the topic. This is why we would prefer to keep the wording.
Lines 48-49: the suggestion is applied.
Best regrads,
Maria Kachamakova
Reviewer 3 Report
In pdf file, highlighted in yellow, authors may notice some missing letters, missing punctuation or incomplete words. In all the references, DOIs must be included according to MDPI style. Otherwise, I have no additional comments on the manuscript.

Author Response
Dear Editor, Dear Reviewer,
The corrections suggested in the PDF file are applied in the manuscript. Thank you for your contribution. The DOI are included in the references.
Best regards,
Maria Kachamakova